# Real-Time Monitoring of Odour Emissions at the Fenceline of a Waste Treatment Plant by Instrumental Odour Monitoring Systems: Focus on Training Methods

**DOI:** 10.3390/s24113506

**Published:** 2024-05-29

**Authors:** Christian Ratti, Carmen Bax, Beatrice Julia Lotesoriere, Laura Capelli

**Affiliations:** Department of Chemistry, Materials and Chemical Engineering “Giulio Natta”, Politecnico di Milano, Piazza Leonardo da Vinci 32, 20133 Milan, Italy; christian.ratti@polimi.it (C.R.); beatricejulia.lotesoriere@polimi.it (B.J.L.); laura.capelli@polimi.it (L.C.)

**Keywords:** e-nose, continuous monitoring, odour measurement, odour concentration, chemical sensors, gas sensing

## Abstract

Waste treatment plants (WTPs) often generate odours that may cause nuisance to citizens living nearby. In general, people are becoming more sensitive to environmental issues, and particularly to odour pollution. Instrumental Odour Monitoring Systems (IOMSs) represent an emerging tool for continuous odour measurement and real-time identification of odour peaks, which can provide useful information about the process operation and indicate the occurrence of anomalous conditions likely to cause odour events in the surrounding territories. This paper describes the implementation of two IOMSs at the fenceline of a WTP, focusing on the definition of a specific experimental protocol and data processing procedure for dealing with the interferences of humidity and temperature affecting sensors’ responses. Different approaches for data processing were compared and the optimal one was selected based on field performance testing. The humidity compensation model developed proved to be effective, bringing the IOMS classification accuracy above 95%. Also, the adoption of a class-specific regression model compared to a global regression model resulted in an odour quantification capability comparable with those of the reference method (i.e., dynamic olfactometry). Lastly, the validated models were used to process the monitoring data over a period of about one year.

## 1. Introduction

Nowadays, odour pollution is one of the main causes of public complaints to local authorities [1]. Several human activities, especially those related to waste treatment and disposal, can lead to the formation of unpleasant odours. Even though odour emissions are, in general, not harmful to human health, they often generate concerns among the people living in the proximity of these sources [2,3]. Therefore, odour is now considered a pollutant, and it has also been included in the Best Available Techniques (BAT) reference document for waste treatment plants (WTPs) among the pollutants to be monitored and controlled from this type of installation [4]. Odour monitoring, which is applied whenever an odour nuisance at a sensitive receptor is expected or has been substantiated, can be carried out using analytical methods or sensorial approaches [5]. Among analytical tools, Instrumental Odour Monitoring Systems (IOMSs), including electronic noses (e-noses), have been recently increasingly studied for air quality monitoring applications [6,7]. Several studies prove that, if properly trained, IOMSs are able to detect the presence or the absence of odours and discriminate between different odour emission sources [8,9]. The first studies describing the application of e-noses for the monitoring of odours at receptors for a limited duration date back to the early 2000s–2010s [10,11]. Recently, some scientific studies have also discussed the possibility of using e-noses for estimating odour concentration [12,13]. For more details on the applicability of e-noses for environmental monitoring, we refer to some very recent review papers on the topic [6,7]. Still, their application as real-time air quality monitoring tools for permanent installations is still far from being state-of-the-art. Nonetheless, in Italy, in the last 2–3 years, there have been some cases in which the Competent Authority has requested a WTP to realize a permanent installation of IOMS with the purpose of continuously analysing the ambient air and providing real-time identification of anomalous odour events. If properly working, this type of installation could act not just as an emission monitoring instrument but could represent an effective tool for odour mitigation, allowing the recognition of the causes of the odour events, thereby enabling the plant operators a prompt and targeted intervention and thus reducing the impact on the neighbourhoods.

As previously mentioned, despite its potentialities, the implementation of IOMS for continuous odour monitoring in real-life applications is still challenging [14], and is mainly hindered by some critical aspects related to prolonged outdoor use in critical conditions. Such critical aspects include sensors’ drift over time and cross-sensitivity to atmospheric variables and non-odorous gases [15,16,17]. Cross-sensitivity towards temperature and humidity is particularly critical for environmental odour monitoring, because such parameters may vary within a rather wide range of values, thereby negatively affecting the sensors’ outputs. As an example, typical relative humidity variations that may occur outdoors range from 20% to 100%, whereby some experimental studies have proven gas sensors typically used in e-nose systems (i.e., MOX sensors) to be sensitive to humidity variations of 1%.

For this reason, different humidity and temperature compensation strategies have been proposed in the literature [18] such as interference compensation models [19,20,21], methods based on separation models for environmental factors [22] and methods based on hardware optimization [23,24]. To the best of our knowledge, such solutions have been studied and implemented in laboratory conditions, while the application of compensation strategies in real scenarios is still very limited. Indeed, laboratory experiments are carried out in a controlled environment, whereas when operating outdoors, variable background conditions, as well as the presence of unknown interferents, may affect sensors’ responses in an unpredictable way compared to what is observed in the lab, thus reducing the effectiveness of the compensation strategies developed in lab conditions. 

In this context, this paper describes the implementation of a continuous odour emission monitoring system consisting of two e-noses at the fenceline of a WTP. Besides describing the experimental procedure involved in training and testing the IOMS in situ, the paper focuses on the development of a specific pre-treatment procedure to be applied to the monitoring data with the purpose of compensating oscillations of the e-nose signals due to temperature (T) and relative humidity (RH) variations. Two approaches, which differ in the definition of the reference baseline, have been investigated, and the efficacy of the two methods was evaluated in terms of classification and quantification performance achieved within specific field verification tests and compared with the ones obtained without applying any correction. 

Furthermore, the paper reports the results of the IOMS monitoring carried out at the WTP fenceline for about one year. The frequency of odour events and the trend of the odour concentration at the monitoring sites were evaluated and correlated with other useful information regarding the plant operation, such as extraordinary maintenance activities, temporary shutdowns of the biogas upgrading section, etc., which could potentially be correlated with increased odour emissions, with the purpose of proving the possibility of using the IOMS network as an effective tool for odour management.

The novelty of this work is not related to the development of new algorithms, but rather to the whole procedure adopted for the realization of an effective odour monitoring system capable of providing accurate and useful information in a complex, real-life scenario.

## 2. Materials and Methods

### 2.1. Plant Description

The plant under investigation is a WTP that converts the organic fraction of municipal solid waste (OFMSW) into biomethane and high-quality compost (Figure 1). After pre-treatments (plastic and sand separation), the OFMSW is mixed with water and transferred to anaerobic digestion tanks. Anaerobic digestion leads to the formation of biogas and digestate. The biogas is then refined to biomethane in an upgrading section, while the digestate is sent to an aerobic treatment section to produce high-quality compost. 

In order to prevent odour emissions, OFMSW storage and treatments are carried out in closed sheds operated under a slight vacuum and equipped with ventilators collecting exhaust air to a biofilter unit, which is the only conveyed emission of the plant. However, fugitive emissions from sheds cannot be completely neglected.

Thus, the following odour sources were considered in the study: the fugitive emissions from the sheds used for the storage and pre-treatment of the OFMSW and from the buildings used for the maturation and storage of compost, fugitive biogas emissions from the digestor and the upgrading section, and the conveyed emissions from the biofilters treating the air sucked from the buildings containing the OFMSW and the compost.

### 2.2. Monitoring System

For this study, as hardware, we used two identical e-noses WT1 (WT1 1179 and WT1 1180) commercialized by Ellona (Tolouse, France) and one weather station Vantage Pro2, commercialized by Davis Instruments Corporation (Hayward, Charlotte, NC, USA). Each WT1 e-nose is equipped with an array of 7 sensors, comprising 4 semiconducting metal oxides (MOX) sensors, 2 electrochemical sensors specific for the detection of hydrogen sulphide (H_2_S) and ammonia (NH_3_) and 1 photoionization detector (PID) calibrated with isobutylene. The instruments are also equipped with 2 sensors for monitoring temperature and relative humidity. The weather station Vantage Pro2 is equipped with a group of sensors including temperature and humidity, rainfall meter and anemometer, which offers the possibility to acquire meteorological data with a frequency of 15 min. All the data acquired by the e-noses and the weather station can be visualized and downloaded through the Ellona web interface for further elaboration.

Data processing was implemented externally from the Ellona software v. 2.4.2., in order to develop the compensation models required by the specific application, as described in the following sections.

The installation points for the e-noses have been defined based on the results of a parametric dispersion modelling study, which was carried out with the purpose of evaluating a preliminary correlation between odour concentration at the plant fenceline and the odour concentration expected at the nearest receptors. From this analysis, the most impacted receptors were identified and the selection of the instruments’ installation points was defined to be representative of the emission plumes reaching the most impacted receptors most frequently.

Based on this preliminary study and other logistic considerations, it was decided to install the e-nose WT1 1180 at the Eastern boundary of the plant in the proximity of the biofilters and the entrance of the shed where the OFMSW is discharged and pre-treated before the anaerobic digestion process. The installation site of the WT1 1180 is also very close to the biogas upgrading section. The e-nose WT1 1179 was installed on the opposite side of the plant (Western boundary), i.e., on the rear of the waste receiving and treatment sheds, but close to the two digesters and the torch.

### 2.3. Electronic Nose Training

#### 2.3.1. Data Acquisition

The main purpose of the e-nose training is to create a reference dataset (i.e., training set, TS) to be used for the implementation of models for classifying and quantifying the air analysed by the instruments during the monitoring phase. To carry this out, dedicated olfactometric campaigns are needed to characterize the plant odour emissions sources. Measurements should be carried out in different periods in order to account for both the intrinsic variability of the sources and the variability associated with the seasonality and different weather conditions. In this case, a total of 70 odour samples were collected directly at the odour sources of the plant as listed in Section 2.1 and analysed by dynamic olfactometry (EN 13725:2022) [25] to assess their odour concentration. Samples were collected over 5 olfactometric campaigns, conducted in different seasons between February 2022 and October 2022, and with different meteorological conditions.

In more detail, odour samples were collected in WTP sheds devoted to OFMSW treatment by means of a mechanical depression pump. Samples of exhaust gas from biofilters were collected by means of a static hood, which allows the isolation of a part of the biofilter surface and conveys the exhaust gas flux through the exit duct where the sample is taken by means of a mechanical depression pump. Finally, samples of fugitive biogas emissions were taken from the gas transfer pipeline from the digestor to the upgrading section, exploiting the overpressure of the gas flow for filling the gas sampling bag. In order to create a TS representative of conditions expected at the monitoring sites, characterized by lower concentrations than those measured at the emission sources, odour samples were diluted with odourless ambient air taken at the plant fenceline when odours were not perceivable by the operators.

After dilution, from the initial 70 samples collected directly at the odour sources, we obtained a total of 240 samples (Table 1), with odour concentrations ranging from about 20 ou_E_/m^3^ to about 1000 ou_E_/m^3^.

In more detail, 196 samples collected during the first four campaigns were used to build the TS, whereas the 44 samples of the fifth campaign were used as external independent test sets for field performance verification, as described in Section 2.4 (Table 2).

Samples were presented to the e-nose directly in the field to account for background conditions at the monitoring sites and prevent bias in the TS. Odourless ambient air samples were also analysed to define the baseline of IOMS signals when no odour could be perceived. Sensor responses accounting for various analyses in the field were registered for the next steps of data processing.

#### 2.3.2. Data Processing

The general scheme of the data processing procedure developed for this study is illustrated in Figure 2. The details of each step will be discussed in the following sub-sections. After an initial pre-treatment aimed at normalizing the data and compensating potential interferences, different features from the sensors’ signal curves were extracted and selected by means of the Boruta algorithm [26]. Selected features were used as input for principal component analysis (PCA) to conduct an exploratory analysis [27,28]. Then, a two-step machine learning model was implemented. In the first phase, a Support Vector Machine (SVM) [29,30] classification model was built to predict the class of odours detected at the fenceline. In the second step, a class-specific regression model based on Support Vector Regression (SVR) [31] was developed to estimate the odour concentration. These algorithms were chosen because they proved to have a better performance on the specific dataset considered in this study compared to other classification algorithms (e.g., Random Forest and KNN) and other regression algorithms (e.g., PLS) [32]. Different approaches were compared to find out the optimal data processing procedure for the specific case, as will be described in the following sub-sections.

##### Pre-Treatment and Normalization of the Sensors’ Resistances

As already mentioned, one of the main issues associated with the application of e-noses for environmental odour monitoring is their cross-sensitivity towards humidity and temperature variations. Such interference cannot be neglected in the case of outdoor installations, where considerable changes in temperature and humidity levels of the ambient air occur because of the day–night cycles, the season, and the variable meteorological conditions. All these variations cause alterations in e-nose sensor resistances comparable to those observed in the case of odour events. Thus, if not properly treated, they are likely to lead to misinterpretation of the monitoring results and overestimation of the odour detections.

For this reason, we decided to explore the possibility of applying a specific pre-treatment aimed at compensating interferences on e-nose sensor responses caused by temperature and humidity variations occurring during the monitoring, which could have been misclassified as odour events.

The proposed pre-treatment involves the normalization of raw sensor signals recorded during the monitoring versus a reference baseline, as reported in (1),
(1)$$y_{norm}=\frac{y-y_{ref}}{y},$$
where *y_norm_* is the normalized value of resistance, *y* is the raw resistance value of the sensor and *y_ref_* is the value of reference baseline. Two approaches for the definition of the reference baseline have been investigated: the first one uses as parameter *y_ref_* as the mean sensor resistance value of the different sensors exposed to odourless ambient air, while the second approach is characterized by the implementation of a variable ‘dynamic’ *y_ref_* obtained through a regression model, which correlates the value of resistance in non-odorous ambient air with the values of absolute humidity (Figure 3).

Figure 3 shows, as an example, how the electrical resistance of one MOX sensor of the e-nose, exposed to odourless air, changes with the moisture content of the analysed air. In more detail, Figure 3 compares the MOX sensor baseline with the absolute humidity, expressed in g/m^3^ and defined by (2),
(2)$$AH=\frac{RH\;P_{s}}{R_{w\;}T\;100\;},$$
where *AH* is the absolute humidity, *RH* is the relative humidity, *T* is the temperature of the air expressed in K, *R_w_* is the specific constant of gases for water vapour (equal to 461.5 J/kg/K), and *P_s_* is the water vapour pressure evaluated using the empiric Equation (3) available on Perry’s Chemical Engineers’ Handbook [33],
(3)$$\ln P_{s}=C1+\frac{C2}{T}+C3\ln T+C4\;T^{C5}$$
where *C*1, *C*2, *C*3, *C*4, and *C*5 are empiric constants specific for the evaluation of the vapour pressure of water and *P* is the pressure expressed in Pa.

The visual inspection of the plot in Figure 3 points out the existence of a decreasing trend of the sensor’s baseline with increasing absolute humidity, thereby suggesting the possibility of finding a correlation between these two variables through a regression model.

The regression model we chose consists of a polynomial regression based on a formula such as the one reported in (4),
(4)$$y_{ref}=a_{1}+a_{2}AH+a_{3}AH^{2}+a_{4}AH^{3}+a_{5}AH^{4}+a_{6}AH^{5}$$
where the *a* coefficients are the empiric constants obtained from the regression model and *AH* is the absolute humidity value evaluated using the temperature and relative humidity registered by the instruments during the analysis. Regression models with different polynomial curves, up to the fifth grade, were tested for each sensor. The choice of the polynomial model was made mainly by a visual check by trying to select the polynomial curve with a lower degree that better approximates the sensor trend (e.g., first-grade polynomial in the case of a linear decreasing trend or third-degree polynomial in the case of an exponential-like trend).

It should be pointed out that this polynomial regression does not claim to have a physical meaning of the sensor’s change in resistance in consideration of the chemical reactions occurring on the sensor’s surface. However, since the interaction mechanism of water with the sensor surface is still not fully understood, especially in the case of complex mixtures with other chemical compounds, thus making mathematical modelling of the physical phenomena hardly applicable, the only possible way of “correcting” the effect of humidity on the sensors is to use an empirical regression based on the interpolation of experimental data obtained from sensor exposure to different humidity levels. It should be further noted that similar approaches (using polynomials) have been already proposed in the scientific literature [34] to correct for humidity and temperature effects on MOX sensors.

##### Feature Extraction and Selection

After normalization of the sensors’ signals, feature extraction [35], calculated on a 5 min basis, was applied on the TS to extract the numerical parameters useful for the discrimination of the different odour emission sources considered. This step requires the implementation of mathematical operations on the sensor curves to obtain a multidimensional vector with informative values to be used as input for classification and quantification. In this case, 10 features were been extracted for each MOX sensor and 1 feature was extracted for each electrochemical sensor and for the PID, for a total of 43 features extracted (Table 3).

To select the best features that entail useful information allowing the discrimination of a certain type of odour, the data processing procedure includes a step of feature selection. Different types of feature selection algorithms can be applied [36,37,38]. Here, we used Boruta [26], which is an algorithm based on the Random Forest classifier allowing us to rank the features based on their contribution to the performance of classification/regression. Compared to other algorithms, which are based on finding the smallest feature set giving the best classification result, Boruta allows us to identify all the features relevant to the classification purposes.

As a result, eleven features were selected for WT1 1180, and thirteen for WT1 1179, respectively. These features comprise the minimum of the resistance (A), the area under the curve and the minimum of the EMA with alpha 0.1 (D,E) [39], the difference between the maximum and minimum of the resistance (F), the minimum of the derivative of the resistance (G) and the maximum reading of the H_2_S electrochemical sensor (L).

##### Exploratory Analysis

On the selected features, a principal component analysis (PCA) was applied to reduce data dimensionality and obtain a graphical visualization of the instrument’s capability to discriminate the different odour classes (Appendix A). Also, the PCA allows us to identify possible odour samples acting as outliers, which are removed from the TS before the implementation of the classification and regression models. In this type of application, outliers are mainly attributable to sampling issues (e.g., contamination of the sampling lines, unpredictable variations in the sampled emission, etc.), which are not always clearly recognizable a priori, and thus are required to be identified through a graphic interpretation of the sample distribution in the so-called influence plot [28].

##### Classification and Regression Models

Not all the samples analysed (Section 2.3.1) were used for the implementation of the classification and regression models. As previously mentioned, the training models were developed using only 196 out of 240 samples, because the remaining 44 were collected later and left out as an external test set to assess the e-nose performance during operation on the field (as described in Section 2.4). Moreover, ca. 10% of the training samples were identified as outliers and removed to optimize the TS. Furthermore, the PCA allowed us to observe similarities between some of the odour classes considered, which made them hardly discriminable. Therefore, the final training model was built based on three classes: ‘Air’, which is representative of the condition of absence of odour; ‘Biogas’, which refers to potential fugitive emissions from the safety valves on the biodigesters; and ‘Organic odour’, which refers to the fugitive emissions from the sheds used for the receiving and pre-treatment of the organic wastes and the building dedicated to the maturation and storage of the compost and the biofilters.

The odour pattern registered by the e-nose is different depending on the odour class. As a consequence, samples belonging to different odour classes with the same odour concentration will provoke a different sensor response pattern, because of the different chemical composition of the odour classes. For this reason, the construction of an effective quantification model should be based on the identification of the odour class, first. Also, the identification of the odour class is of particular importance for the plant managers, because it enables the identification of the causes of the odour events detected at the fenceline.

The models implemented for classification and regression are based on the non-linear SVM (Support Vector Machine) with a Gaussian radial basis function (RBF) kernel [30]. As mentioned in the previous paragraphs, the score of the data processed by PCA is taken and used as input for the implementation of the classification models, whose outputs define the odour class of the gas analysed. In particular, for both e-noses, the first three principal components account for about 90% of the explained variance and so they were selected as input of the models.

In more detail, three classification models deriving from different data analysis procedures were compared: (1) Model CA—analysis of the raw sensor resistance without any pre-treatment; (2) Model CB—analysis of the sensor resistance after normalization of the signal as reported in Equation (1), with the parameter *y_ref_* fixed at the general sensor resistance value of the different sensors exposed to odourless ambient air; (3) Model CC—analysis of the sensors resistance after normalization with absolute humidity compensation of the signal as reported in Equation (1), with a variable *y_ref_* evaluated with the regression model expressed in Equation (4) at the absolute humidity measured during the analysis time window. The models obtained were optimized by means of internal 10 k-fold cross validation and to determine their performance an independent dataset was used for testing [40].

Regarding odour quantification, two different strategies were compared: the first approach (Model QA) is based on the application of a global regression model trained using a training set comprising all samples without differentiation of the odour class; instead, the second approach relies on a classification of the odour analysed prior to regression (Model QB). In this case, different regression models are built based on the number of odour classes defined in the TS (Figure 2). The aim of the comparison is to verify the assumption that the classification of the odour prior to its quantification can improve the accuracy of the odour concentration prediction.

The evaluation of the regression models’ performance was conducted by comparing the odour concentration predicted on the independent test set with the concentration obtained by the reference method, i.e., dynamic olfactometry, and then by applying a statistical analysis based on the Bland–Altman (B&A) model [41,42].

### 2.4. Performance Verification in the Field

During the monitoring phase, it is important to carry out specific field tests to evaluate the reliability of the instruments’ detections. Field performance testing is also one of the requirements of the recently published Italian technical norm UNI 11761:2023 [43] on instrumental odour measurement. In this case, sampling for field performance testing was carried out as for the training phase, but in a separate campaign after the definition of the training models: gas samples were withdrawn from the different odour emission sources considered, analysed by dynamic olfactometry (in accordance with EN 13725:2022), diluted, and then analysed by the e-noses to verify the instrument accuracy in recognizing the odour class and quantifying the odour concentration.

The odour classification capability was assessed in terms of the Matthews Correlation Coefficient (MCC), which is a better indicator compared to the global accuracy when the classes of interest are not balanced [44].

The Bland–Altman method was then applied to evaluate the odour quantification capability and compare the odour estimates by the e-noses with the reference method. This approach was chosen because dynamic olfactometry has a high measurement uncertainty (generally considered at about 50% [13]), which makes a comparison using traditional calibration methods based on the correlation coefficient (r) unsuitable [45]. For this reason, it is necessary to adopt “model comparison” methods, which enable us to compare outputs obtained from methods having comparable levels of uncertainty. The Bland–Altman method has been recently proposed as a suitable approach to provide quantitative information about the capability of IOMS to estimate odour concentrations [13].

To properly apply the Bland–Altman method, it is first necessary to check whether the differences in the values calculated with the two methods are normally distributed. This can be evaluated, for example, by applying a Shapiro–Wilk test [46] and verifying the resulting *p*-value. If the *p*-value is higher than 0.05, then the condition of normality cannot be rejected. It should be considered that, when dealing with odour concentration values, the agreement should be evaluated by comparing the differences in their logarithms.

The application of the Bland–Altman method provides the bias, which is the mean of the differences, and the Limits of Agreement (Lower LoA and Upper LoA), which define a range within which 95% of the differences between one measurement and the other are included.

Performance testing was mainly conducted with the purpose of comparing the different compensation strategies and the different regression models developed, finding the most suitable approach to develop a robust and accurate quantification model.

### 2.5. Monitoring Data Evaluation

After installation, the e-noses analysed the ambient air continuously, and the sensor resistance values were registered every 10 s. The monitoring phase lasted for about 1 year, resulting in a monitoring dataset of 73,685 observations.

These data were processed according to the data processing procedure described in Section 2.3.2. In more detail, data acquired continuously were divided into time intervals of 5 min to extract features from the sensor curves and process them with the classification and regression models developed. This means that the monitoring system produces an output every 5 min in terms of odour class detected and estimated odour concentration.

Monitoring data were analysed with the purpose of obtaining information about the detection of odours at the fenceline and their origin. Data were analysed first with the purpose of evaluating the odour impact in terms of the ratio between all the ‘odour’ detections (i.e., measurements classified differently than ‘air’) and the total number of measurements. Then, the relative detections of the different odour classes and the odour concentration trends over time were evaluated as well, with the purpose of identifying the most critical odour classes causing the majority and/or more intense odour events.

## 3. Results

### 3.1. Field Performance Testing

#### 3.1.1. Odour Classification Performances

The different absolute humidity compensation models were evaluated by analysing the odour classification performances of the e-noses WT1 1179 and WT1 1180 when applying models.

The models were first validated (by internal cross-validation) and then tested by using an independent dataset as the test set. The confusion matrices resulting both from the validation and testing were assessed to compare the models’ performances.

The results, expressed in terms of MCC values obtained with the different classification models for each electronic nose, are reported in Table 4.

It is possible to observe similar behaviours for both e-noses: the MCC values relevant to the models implemented without signal normalization (Model CA) perform the worst, while the models obtained after normalization and compensation with respect to the absolute humidity (Model CC) provide the best results. For the e-nose WT1 1179, the result performances obtained with the different models are not huge, with a minimum value of MCC obtained on the independent test set equal to 0.85 for Model CA and a maximum of 0.95 for Model CC. The e-nose WT1 1180, instead, shows greater variability in the performances obtained depending on the data processing procedure applied, ranging from 0.69 to 1 on the independent test set for Model CA and Model CC, respectively.

The results obtained highlight the importance of considering the interference of humidity and temperature in the response of the sensors in order to obtain reliable results in terms of odour class recognition.

The classification Model CC was further tested with an independent dataset. The results of this validation, reported as confusion matrixes in Figure 4a,b, show that, after the application of a suitable model for compensating the sensor responses considering the instantaneous absolute humidity value, the WT1 1179 and WT1 1180 are capable of properly discriminating the different odour classes with accuracy indexes of 97% [95% CI: 85–99%] and 100% [95% CI: 88–100%], respectively.

#### 3.1.2. Quantification Performances

Field testing was also applied to evaluate the capability of the e-noses to estimate the odour concentration, thereby comparing the odour concentration estimates from the instruments with the odour concentration values measured by dynamic olfactometry (according to EN 13725:2022).

The comparison was made by applying the Bland–Altman analysis, which can be used to estimate the agreement between two different methods [41,42]. The Shapiro–Wilk test was applied to both e-noses and for both models (Model QA and Model QB) obtaining a *p*-value higher than 0.05 in every case analysed, meaning that the condition of normal distribution of the differences cannot be rejected. Figure 5 reports the Bland–Altman plots relevant to Models QA and QB for the two e-noses, WT1 1179 and WT1 1180, respectively. The *x*-axis reports the mean of the logarithm of the odour concentrations, while the *y*-axis reports the differences in the logarithm of the odour concentrations. The plots further report the field measurements (black points), the bias (green dash-dot line), the Upper Limit of Agreement (LoA, blue dash-dot line), the Lower LoA (pink dash-dot line) and their 95% confidence intervals (delimited by the corresponding coloured dot lines). By looking at the plots in more detail (Figure 5), both Models QA and QB have their equality line (i.e., the line where the differences are equal to zero) contained in the 95% confidence interval of the bias, meaning that the bias is not significant.

For an easier and more immediate comparison of the two models, the LoA is also expressed in Table 5 and Table 6 in terms of multiplicative factors.

For the e-nose WT1 1179, the Lower and Upper LoA for Model QA are 0.13× and 4.27×, respectively, instead for Model QB they are equal to 0.24× and 2.6×. For the e-nose WT1 1180, these values are 0.24× and 3.13× for Model QA and 0.28× and 2.39× for Model QB. The outcomes of this analysis highlight a difference in the level of agreement between Models QA and QB for each e-nose. In more detail, Model QB results provide, for both e-noses, better estimates compared to the odour concentration values measured by the reference method.

To further compare the different models, we tried to assess the e-nose quantification performance adopting the same criteria for accuracy and for intermediate precision indicated in EN 13725:2022 for dynamic olfactometry.

According to the EN, to satisfy the accuracy criterion, the logarithm of the bias needs to be lower than 0.217 (Equation (5)), while the intermediate precision criteria are satisfied if the standard deviation(s) of the observations, in a non-logarithm form, multiplied by 1.96, is lower than 3 (Equation (6)), meaning that the factor that expresses the difference between two consecutive odour concentration measurements will not be larger than a factor of 3 in 95% of the cases.
(5)$$\left|{log}_{10}bias\right|<0.217$$
(6)1.96∗s<3

The same criteria were applied for the evaluation of the e-nose estimates, and the results are reported in Table 7. The accuracy criterion is satisfied by all the models for both e-noses since the absolute value of the logarithm of the bias is always lower than 0.217. On the other hand, the intermediate precision criterion is not satisfied by any of the models developed. Even if the criterion is not respected, comparing Models QA and QB, it is possible to observe that, for both electronic noses WT1 1179 and WT1 1180, Model QB is the one performing better, giving a factor closer to the limit value of 3. Based on these considerations, it was decided to adopt Model QB as a quantification model to predict the odour concentration of the ambient air analysed by the instruments at the plant fenceline.

The final data processing procedure applied and the different approaches considered have been resumed in Figure 6.

### 3.2. Monitoring Results

The signals registered by both e-noses from February 2022 to November 2022 were processed with the aim of obtaining a real-time qualitative and quantitative characterization of the ambient air at the plant fenceline. Based on the results discussed in the previous sections, we decided to adopt Model CC to compensate for humidity and assess the odour class, whereas the estimation of the odour concentration was obtained relying on Model QB (Figure 6).

The monitoring results have been then elaborated to assess the frequency of detection of the different odour classes considered at the instruments’ installation sites.

Figure 7 reports the odour detection frequencies for both e-noses, WT1 1179 and WT1 1180. It can be observed that the WT1 1180 detected odour for about 11% of the monitoring period with a quasi-equal distribution between the two odour classes ‘Biogas’ and ‘Organic odour’ (5% and 6%, respectively). Instead, the WT1 1179 detected odour for about 7% of the monitoring period, with the odour events belonging principally to the ‘Organic odour’ class, which was recognized 6% of the time, compared to the odour class of ‘Biogas’, which was detected only ca. 1% of the time.

These differences can be explained by considering the different positions of the two e-noses. The WT1 1180 is installed in a more ‘critical’ point that allows the detection of odour emissions from the biofilters and the sheds where the OFMSW is stored and processed; moreover, the WT1 1180 is located also downwind the upgrading section, thus enabling the detection of fugitive emissions of biogas from that area, which turned out to be not negligible. Instead, the WT1 1179 is installed close to the biodigester and the torch, from which fugitive biogas emissions seem to be more limited.

The outputs of the regression models for WT1 1179 and WT1 1180 have been reported in Figure 8 and Figure 9, showing the predicted odour concentration for the entire monitoring period. Looking at the results for the WT1 1179, the e-nose detected more frequent odour events attributed to the ‘Organic odour’ class; odour events related to ‘Biogas’ are rare, but they result in relatively high odour concentration peaks (up to ca. 1000 [oue_eq_/m^3^]). Conversely, the WT1 1180 detected odours attributed to the two classes ‘Organic odour’ and ‘Biogas’ with similar frequencies, and the relevant odour concentrations reached up to ca. 500 [oue_eq_/m^3^].

What is particularly interesting from the point of view of the plant management is that the odour events recorded by the two e-noses, and especially those associated with high odour concentrations, could be often related to specific malfunctioning events or anomalous operating conditions of the plant. As an example, the increase in the frequency of the odour events registered by the e-nose WT1 1180 in October and November coincides with a malfunctioning of the ventilation system of the OFMSW storage buildings, which resulted in increased fugitive odorous emissions, because of the reduced capacity of keeping the shed’s interior at a negative pressure. Moreover, the high concentration peaks of biogas detected by the e-nose WT1 1179 in October could be identified as related to extraordinary maintenance operations carried out on the pipelines transporting the biogas from the biodigester to the upgrading section.

## 4. Conclusions

The present study discusses the possibility of using e-noses to continuously monitor the odour emissions at the fenceline of a WTP by detecting the presence of odours, recognizing their provenance (odour source) and estimating the odour concentration. In more detail, this study focuses on the instrument training methods, thereby comparing the application of different training models.

First, the training involved extensive field data acquisition: 196 samples of known odour quality and concentration collected at the plant’s main odour sources over four different olfactometric campaigns conducted in different seasons were analysed by the instruments, opportunely diluted to reproduce odour concentration levels that are likely to be detected at the plant fenceline. Furthermore, other 44 independent samples collected on a fifth campaign were used for testing.

Then, the work involved the implementation and the comparison of different models for the compensation of ambient air humidity and temperature, which are known as main interferents in the application of e-noses for environmental monitoring. Results show that the implementation of a suitable corrective model of sensor responses to compensate for changes in absolute humidity of the sample significantly improves the odour classification performances, passing from accuracies of ca. 70% to 100%. These observations further highlight the importance of including of a suitable compensation strategy whenever the e-nose is used outdoors, with variable atmospheric conditions.

Finally, in this study, we also studied and compared different models for the estimation of the odour concentration. In more detail, we wanted to confirm the hypothesis that a ‘double-step’ model, including first the identification of the odour class and then applying a regression model specific for the recognized odour class, would be more effective than a more simplified regression model without prior identification of the odour class. Indeed, the ‘double-step’ model proved to provide more accurate estimations, resulting in reduced factors for the Limits of Agreement evaluated by the Bland–Altman method. Moreover, quantification performances achieved by the trained electronic noses turned out to be comparable with the quality criteria fixed by EN 13725:2022 for the reference method, i.e., dynamic olfactometry.

As a conclusion, the study proves that, if properly trained, by including a suitable signal pre-treatment method for compensating atmospheric variables and by implementing specific quantification models based on a preliminary identification of the odour class for regression, e-noses can be very effective for environmental odour monitoring, achieving classification accuracies above 97% and quantification performances comparable to those considered acceptable by the reference standard for dynamic olfactometry.

Regarding the final application, the most important result consists of the realization of an effective odour emission monitoring system, enabling the real-time detection of odour events and the identification of their sources. This is a key feature for plant managers for identifying the possible causes of anomalous odour emissions and enabling prompt intervention, thus reducing the odour impact on the neighbourhoods.

Future studies should focus on the optimization of the classification and quantification models developed on a longer monitoring period, thereby accounting for the well-known problem of sensor drift over time and developing specific strategies to counteract it.

## Figures and Tables

**Figure 1 sensors-24-03506-f001:**
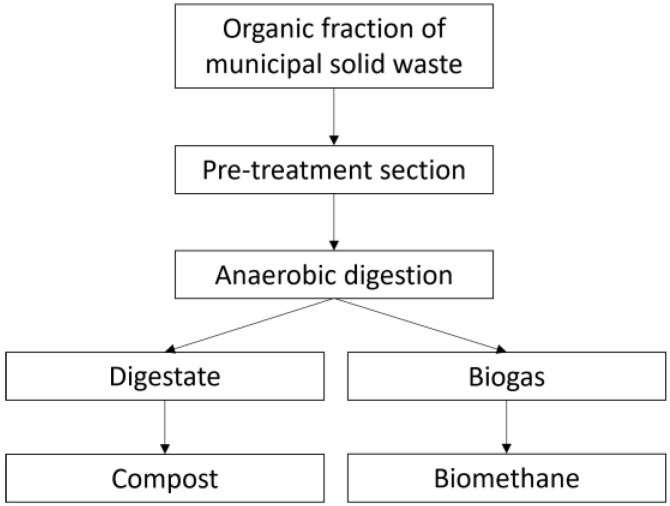
Plant operation scheme.

**Figure 2 sensors-24-03506-f002:**
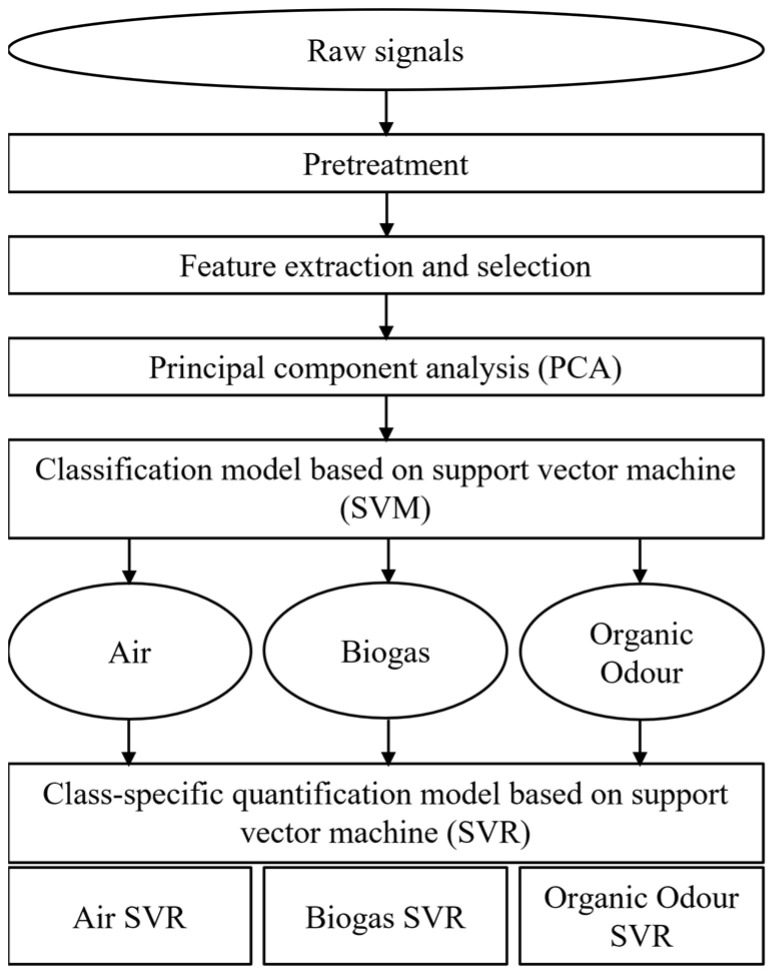
Data processing scheme resuming the different steps applied.

**Figure 3 sensors-24-03506-f003:**
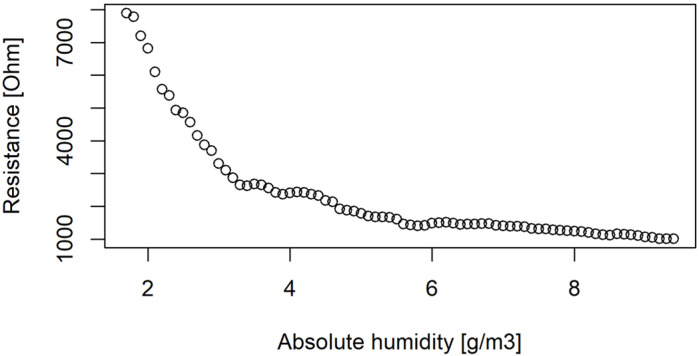
Example of sensor resistance variation exposed to non-odorous ambient air with different values of absolute humidity. The *x*-axis reports the value of absolute humidity in g/m^3^, while the *y*-axis reports the resistance in Ohm.

**Figure 4 sensors-24-03506-f004:**
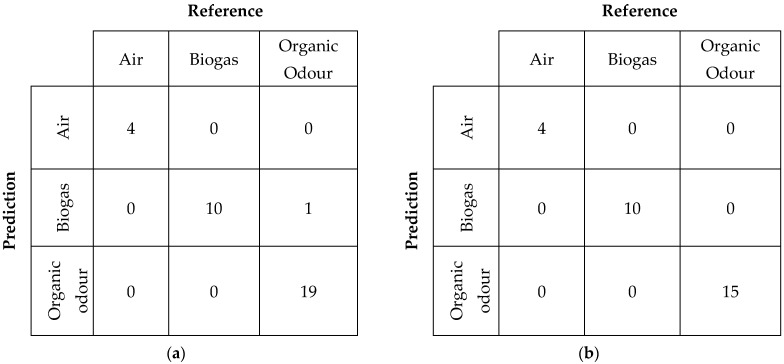
(**a**) WT1 1179 confusion matrix for odour classification resulting from testing with an independent dataset; (**b**) WT1 1180 confusion matrix for odour classification resulting from testing with an independent dataset.

**Figure 5 sensors-24-03506-f005:**
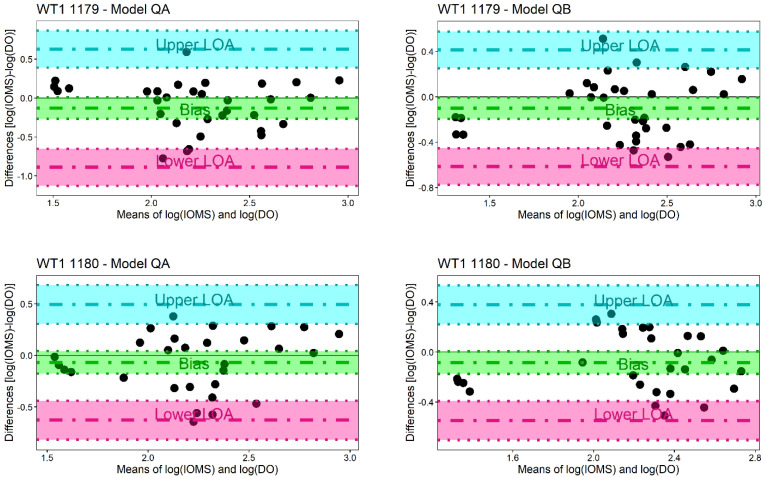
Bland–Altman plots obtained by applying the quantification Model QA and the quantification Model QB on the estimates of the electronic nose WT1 1179 and WT1 1180. In both plots, the bias is reported as a green dash-dot line, the lower limit of agreement as pink dash-dot line and the upper limit of agreement as blue dash-dot line. Each parameter is reported together with the corresponding 95% confidence interval, represented by the dotted lines of the same colour. The *x*-axis reports the mean of the logarithm of the concentration evaluated with the quantification model (IOMS) and the dynamic olfactometry (DO), while the *y*-axis reports their difference.

**Figure 6 sensors-24-03506-f006:**
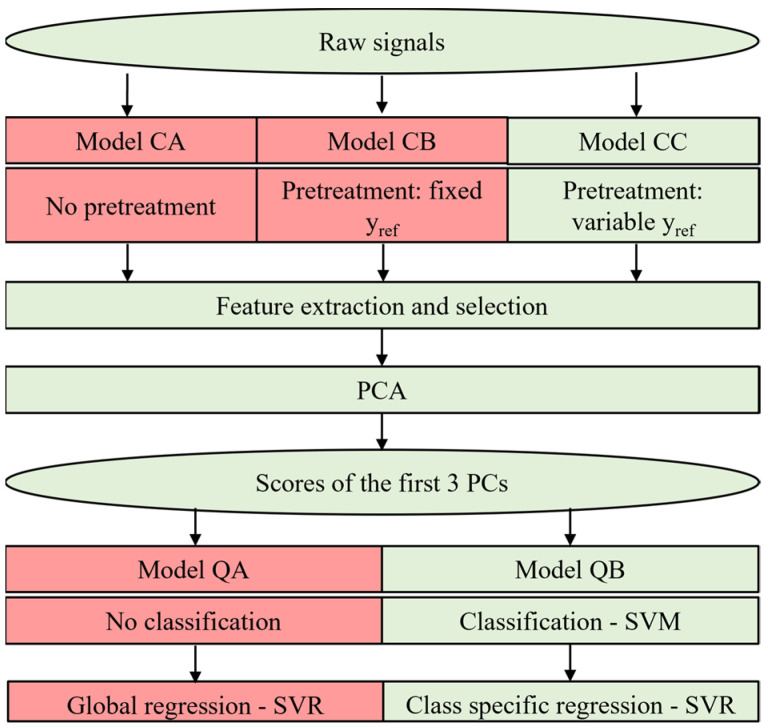
Detailed data processing scheme. The final data path flow are highlighted in green, while the discarded model approaches are coloured in red.

**Figure 7 sensors-24-03506-f007:**
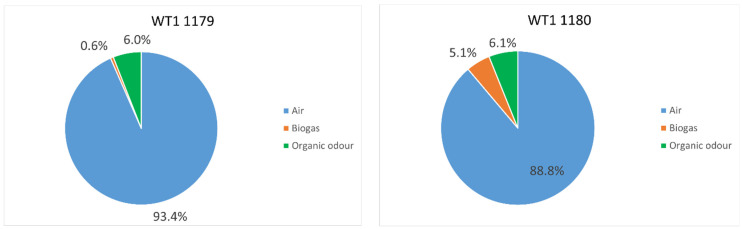
Odour detection frequency between the months of February and November 2022 for the electronic nose WT1 1179 and WT1 1180.

**Figure 8 sensors-24-03506-f008:**
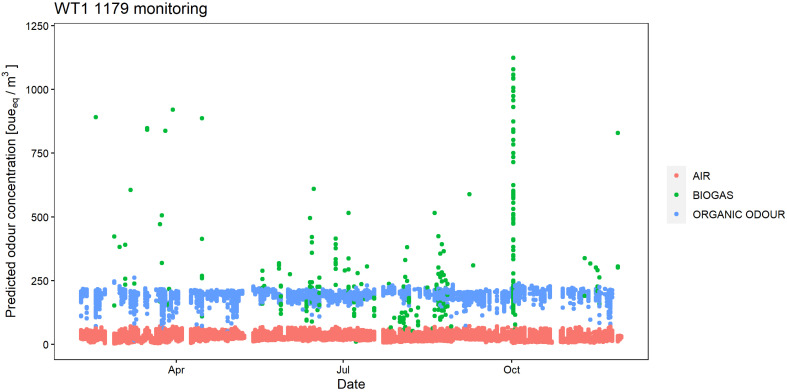
Odour class and relative odour concentration resulting from the ambient air analysis by the electronic nose WT1 1179 at the plant fenceline from February 2022 to November 2022.

**Figure 9 sensors-24-03506-f009:**
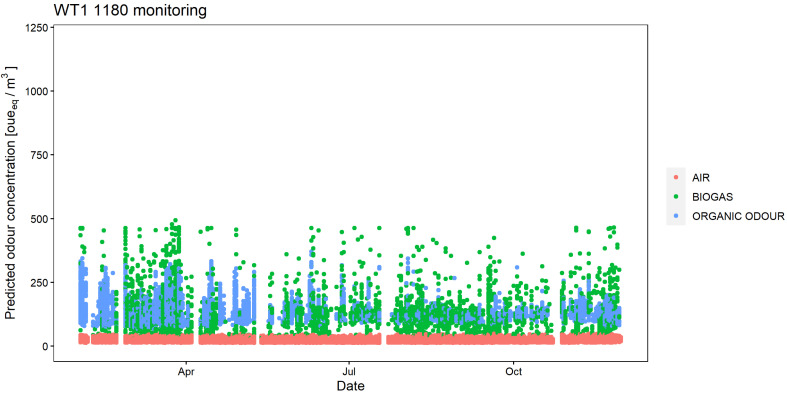
Odour class and relative odour concentration resulting from the ambient air analysis by the electronic nose WT1 1180 at the plant fenceline from February 2022 to November 2022.

**Table 1 sensors-24-03506-t001:** Odour class and concentration range of the samples analysed by the electronic noses for the training and for the field tests for performance verification.

Class	Number of Diluted Samples Analysed	Odour Concentration Range [ou_E_/m^3^]
Air	33	20–50
Biogas	65	22–990
Biofilter	49	32–860
Sheds	93	27–950

**Table 2 sensors-24-03506-t002:** List of all the samples collected during the different campaigns and their division between training set and field test set.

	Training Set				Test Set
	22 February	22 April	22 May	22 July	22 October
Number of independent samples collected at the source	13	17	6	15	19
Number of diluted samples	47	65	25	59	44

**Table 3 sensors-24-03506-t003:** List of the features extracted.

SMOX Sensors	Feature Name	Description
	A	Minimum of the resistance
	B	Exponential moving average (EMA)—area under the curve—alpha 0.05
	C	Exponential moving average (EMA)—minimum—alpha 0.05
	D	Exponential moving average (EMA)—area under the curve—alpha 0.1
	E	Exponential moving average (EMA)—minimum—alpha 0.1
	F	Difference between maximum and minimum of the resistance
	G	Minimum of the derivative of the resistance
	H	Area under the curve
	I	Mean of the resistance
**Electrochemical Sensor and PID**	**Feature Name**	**Description**
	L	Maximum of the sensor reading

**Table 4 sensors-24-03506-t004:** Comparison of the performance of different humidity compensation models for the electronic nose WT1 1179 and WT1 1180. The ‘Validation’ MCC is referred to the Matthews Correlation Coefficient evaluated on cross-validation, while the ‘Test’ MCC is referred to the same coefficient calculated on the independent test dataset used for testing.

	WT1 1179	
Model	‘Validation’ MCC	‘Test’ MCC
Model CA	0.86	0.85
Model CB	0.91	0.9
Model CC	0.91	0.95
	**WT1 1180**	
**Model**	**‘Validation’ MCC**	**‘Test’ MCC**
Model CA	0.55	0.69
Model CB	0.85	0.77
Model CC	0.88	1

**Table 5 sensors-24-03506-t005:** Parameters of Bland–Altman model applied on Model QA and QB for the e-nose WT1 1179.

WT1 1179	Model QA		Model QB	
	Logarithmic Differences	Multiplicative Factors	Logarithmic Differences	Multiplicative Factors
Bias	−0.13		−0.1	
Upper LoA	0.63	4.27×	0.41	2.6×
Lower LoA	−0.89	0.13×	−0.61	0.24×

**Table 6 sensors-24-03506-t006:** Parameters of Bland–Altman model applied on Model QA and QB for the e-nose WT1 1180.

WT1 1180	Model QA		Model QB	
	Logarithmic Differences	Multiplicative Factors	Logarithmic Differences	Multiplicative Factors
Bias	−0.07		−0.08	
Upper LoA	0.5	3.13×	0.38	2.39×
Lower LoA	−0.63	0.24×	−0.55	0.28×

**Table 7 sensors-24-03506-t007:** Results obtained from the Bland–Altman analysis for the quantification Model QA and QB applied on both electronic noses. For each model, the absolute value of the logarithm of the bias, the standard deviation of the difference in the measurements and the result of Equation (6) to check the compatibility with the intermediate precision criteria are reported.

Model	|*log*_10_(*bias*)|	Standard Deviation	Intermediate Precision Criterion
WT1 1179—Model QA	0.13	2.41	4.72
WT1 1179—Model QB	0.1	1.82	3.57
WT1 1180—Model QA	0.07	1.91	3.74
WT1 1180—Model QB	0.085	1.71	3.35

## Data Availability

Data are confidential.

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
