# Peer review of "Real-Time Monitoring of Odour Emissions at the Fenceline of a Waste Treatment Plant by Instrumental Odour Monitoring Systems: Focus on Training Methods"

_sensors, 2024, doi:10.3390/s24113506_

Round 1

Reviewer 1 Report

Comments and Suggestions for Authors

The manuscript presents the use of two IOMS systems for odor detection at the fenceline of a waste treatment plant. Different approaches to compensate for changes in T and RH have been studied and machine learning model consisting of 2 stages is suggested: first: classification of the odor event, second: regression model to determine the intensity.

The manuscript is written very well and clear, relevant works are cited in the state of the art and the presented methodology and results are discussed comprehensively.

I have a few remarks that should be considered prior to publication:

·         Title: the title mentions “electronic noses” but in the abstract and in the introduction the terminology is “IOMS” used which should be preferred. In fact: “e-nose” is not mentioned at all in the abstract. I suggest to change the title to IOMS and use “e-nose” as a key word

·         Abstract: In my opinion the term “IOMS network” is misleading:

o   just to IOMS systems (e-noses) are independently used at the fenceline. A network would share information among each other, and I would expect more than 2 systems to form a network

o   the comparison of ML approaches is not mentioned.

o   The abstract needs to be improved and better tailored to the content of the manuscript.

·         There are many reference errors (Reference source not found)

·         Section 2.2.: “..equipped with 7 sensor arrays…” Are there 7 arrays or 1 array with 7 sensors? Please rephrase the sentence

·         The abbreviation “TS” is not introduced

·         Unclear how the TS looks like. I understood that

o   70 odor samples were taken within the first 4 campaigns. These samples were diluted so that in total 240 are available. From these 240 only 196 are used for building the TS. Correct?

o   Then, there are another 44 samples (not-diluted) from a fifth campaign and used for testing. -> This is really good to have independent data!

·         Section 2.3.2: When first reading the manuscript I got lost / I thought that information is missing. The details of all the signal processing steps come in sub-sections. Please add a sentence at the beginning of of section 2.3.2 saying that all steps will be discussed in the following sub-sections

·         Please make clear that there are several approaches that will be discussed in the following

·         Please present the PCA score plot, maybe in the appendix

·         How many principal components (PCs) are used as input to the SVM? How have they been selected?

·         Fig. 2: I suggest to change the last block into four blocks to highlight that the SVR are dependent on the classification of the SVM

·         Headline section 2.3.2.1 “..electronic signal”: sensor signal? Please rephrase

·         Fig 3: It seems that five polynomial fits were done. Please provide importance scores (R^2, RMSE  or MAE, etc.)

·         Which fit was used? End of section 2.3.2.1 it says “… one that fits best was used” How do you select the best? Which one is the best?

·         Section 2.3.2.2: Which features were used? There is reference  given which is a review about feature selection for e-noses. Which features have you used?

·         For feature selection a multi-variate approach was used with the classification result as a selection parameter. That is good. However, I think that the selected features also depend on the number of used (and selected) principal components? So, it is an iterative process in which different feature sets are selected for different number of PCs to be used as input to the SVM?!!
raw signal -> features -> selected features -> PCA -> SVM

·         Why is the PID sensor not considered? In line 240/241 it says that the MOS sensors and the ECs were used.

·         The terms “validation” and “testing” are mixed also I get what you were tying to say. From a machine learning point of view validation is done on the training set which is split into several partitions in case of k-fold cross validation. The validation is used to tune hyperparameters and to check for overfitting
Then in a second step, an independent dataset (test set) is used to test the overall performance of the model.

Then, the terms “internal” and “external” are introduced. I think that can be improved

·         I understood that for testing 44 samples from an independent campaign were used which has not been used at all in the model building step. Correct? That would be  the best

·         In general, the presentation of the signal processing steps and the selection of the different models could be more clear

·         The two IOMS systems have different performance characteristics in the model building. Does that influence the results in e.g. Fig 7?

·         Fig. 8 and 9:

o   the x-labels are in Italian

o   y-axis scale: comparison of the two plots would be easier, if the axis are the same.

·          

Reviewer 2 Report

Comments and Suggestions for Authors

This work describes a data processing protocol using IOMS network at the fenceline of a WTP for real-time monitoring the odour emissions. The results showed high accuracy and were able to compensate for the interference that generally exists from humidity and temperature.

The research was carried out meticulously in great detail and in very valuable real-life scenarios that can give the audience many good insights.

However, there are a couple of technical points to be addressed before being considered for publication. 

Note that there are many typos and errors of the current manuscript. I have tried my best to get them out. But please proofread more on your revision.

Line 30 - what is BAT? Spell it out.

Line 36-40 - the authors list a couple of existing methods without describing quantitative metrics. Please list them to justify their innovation in comparison.

Line 51-58 - how bad the current situation is for IOMS implementation against moisture and temperatures? Please be specific. The description in the introduction has been too handwaving. 

Line 59-64 - the authors claimed that the limitations exist for studies done in the lab. Please discuss the discrepancy that this can cause such that their actual real-life test is more justified to be superior to other research. 

In the introduction, please elaborate a bit on the state-of-art e-nose for monitoring odours. What are the major gases and what kind of sensors are under implementation? This serves as a context for the later introduction of their adoption of the e-nose design.

Line 87 - fix the reference "Error! Reference source not found.."

Line 134 - "training set". T and S should not be capitalized.

Line 157, 161, 173, 261, 374, 375 - "Error! Reference source not found" fix it.

Line 173 - The authors should elaborate more to justify the adoption of PCA, SVM, and SVR. Why not other algorithms? 

Figure 3 is almost pointless - a higher polynomial can fit the data - and everyone knows that. There is no obvious point to show a figure like this here. Please justify.

Line 231 - "Regression models with different polynomial curve, up to the fifth grade, were tested for each sensor, and the one that best fits the experimental data was used for the calculation of the reference baseline" - similar to my last comment, if there is no physical meaning behind a fit, it is pointless to use a high-order polynomial to fit the data. Everything can be fitted with high orders. Please justify.

Line 244-246 - a sentence or two for describing the advantages of the method chosen can be helpful. 

Tables 2 and 3 should be plotted together with necessary labelling to emphasize the impact of compensation models.

The same is applied to Figures 5 and 6. The same figure captions are taking up too much space for the paper. 

Figure 8 and 9 - X axis is not in English.

Comments on the Quality of English Language

Please proofread your manuscript - there are so many formatting errors.

Reviewer 3 Report

Comments and Suggestions for Authors

In this paper, authors present a study about the odor monitoring at a Waste Treatment Plant. The subject of this research is current and interesting even if many studies have already been carried out about using electronic noses for online odor monitoring.

Two commercial E-noses are placed at the fence line of the plant. After sensor response normalization by the means of humidity and temperature compensation, a classification and then a quantification model are determined. Data collected online about one year are then estimated using these models.

In my point of view, the submission needs an important revision to be worthy of publication in Sensors. The main reasons for this opinion are:

1-    No real novelties or innovations can be found in this study.
In fact, the pretreatment and data analysis techniques used in this study are well known, and involve nothing new.

2-      The main expected result, i.e. online odor quantification, is not clearly discussed.
It is therefore not possible to assess the performance of the whole system (measurement protocol, pretreatment, classification and quantification model).

3-      One of the author’s conclusion is: “The frequency of odor events was correlated with other useful plant information”.
I'm not convinced that this correlation has been demonstrated in the paper, it is only hinted at. It would be necessary to present, at least within some data or period range, the correlation between odor intensities and a particular plant activity.

4-      Some results are given in the text without any justification or explanation.
That is certainly due to a lack of references to previous work by the authors.

For examples:

Data Processing Procedure for Real-Time Odour Concentration Estimation at Industrial Plant Fenceline by Sensor-Based Tools Published in Eurosensors, March 2024

* Realisation of a Multi-Sensor System for Real-Time Monitoring of Odour Emissions at a Waste Treatment Plant Environmental Science Engineering, 2022

The following points also need clarification:

·           Are the two commercial E-noses TW1 (1179 and 1180) identical?

·           Details about the Data collection (observation and features):

o    Campaign dates for training and testing

o    Line 240 : lack of precision concerning the selected features (10 features for SMOX, 1 feature for each electrochemical sensor, and I suppose, one feature for the PID sensor (which is not specified in the section).

o    Utilization of Boruta algorithm for feature selection:
- no information is given on the features selected and their number
- if this part concerns a previous publication that you have made, please give its reference.

·           The use of PCA

o    (line 280)”The score of the data processed by PCA is taken and used as input for the implementation of the classification models What this clearly the meaning of this sentence.
If the first components are used instead of the original features in the classification and quantification models, how this affects the features of the measurements to be estimated.

o    About the outliers: what is the percentage of the outlier’s data? Can you explain the presence of them?

o    Please give a PCA score plot relevant to training set, to see the odor classification cluster.

·           Please explain in section “monitoring data evaluation”, the data which collected every 10 seconds are transformed on a data to be evaluated each 5 minutes.

Round 2

Reviewer 1 Report

Comments and Suggestions for Authors

Thank you for the revision. Good job!

Author Response

we thank the Reviewer!

Reviewer 2 Report

Comments and Suggestions for Authors

Most of the comments were addressed by the authors. However, there is one crucial problem still remaining - 

On response is inadequate after revision and it is very serious a problem scientifically -

They wrote "The choice of the polyno-251 mial model was done mainly by visual check by trying to select the polynomial curve 252 with lower degree that better approximates the sensor trend (e.g., first grade polynomial 253 in case of linear decreasing trend or third degree polynomial in case of exponential-like 254 trend)."

As I mentioned in my previous comment, there is no physical meaning in a polynomial fit. If yes, please justify, otherwise don't try to make a fit, everyone can fit any data using polynomial functions. What does this function tell readers about the data? A mathematical fit means nothing.

Author Response

We totally agree with the Reviewer comment. Indeed, the polynomial fit probably how has no physical meaning.

However, it should be considered that the interaction mechanism of water with the sensor surface is still not fully understood, especially in the case of mixtures with other chemical compounds, which makes a mathematical modelling with a physical meaning of the sensor’s change of resistance in consideration of the chemical reactions happening on the sensor’s surface actually impossible.

In consideration of this impossibility to effectively understand the physical mechanism underlying the humidity effect, the only possibility to “correct” for such effect is to start from experimental data resulting from sensor exposure to different humidity levels, and try to interpolate these data with a reasonable function enabling correction on another set of data.

Our intention is not to provide the reader with a better understanding of the sensor behaviour, but just making it work in a complex environment with variable humidity.

It should be further noted that similar approaches (using polynomials) have been already proposed in the scientific literature (e.g., Abdullah et al., Sensors 2022, doi: 10.3390/s22093301) to correct for humidity and temperature effects on MOX sensors.

Besides, it is true that we have used part of the experimental data to build our polynomial fit to correct the sensors’ responses, but it is also true that the effectiveness of this fit was later validated with other experimental data and with the independent samples used for testing, which were not at all included in the construction of the correction model.

Thus, even though completely agreeing with the reviewer about the fact that a correction model starting with a better understanding of its physical meaning would be desirable, due to the current impossibility of building such a model, we hope that he might partially re-considering his opinion about the empirical approach we adopted for correction.

Reviewer 3 Report

Comments and Suggestions for Authors

Thank you to the authors for the revision made to their article as well as for the additional information. The submission may be published in its present form.

Author Response

We thank the Reviewer!

Round 3

Reviewer 2 Report

Comments and Suggestions for Authors

Understood your points. In this case, please state explicitly the limitations of the current model emphasizing such issues with the relevant references. I need to see them in the next final revision. Or, I prefer you don't do a meaningless polynomial fit and leave it open for future work when a model might be readily ready. Either way should work. The key point I want to bring up is that we, as researchers, must be rigorous in our interpretations and not leave loose ends.

Author Response

We would like to thank the Reviewer for the time she/he has dedicated to revise our manuscript. 

In the last revision of the paper, we have stated explicitly the limitations of the current model, including Abdullah, A. N. et al. Correction Model for Metal Oxide Sensor Drift Caused by Ambient Temperature and Humidity. Sensors 22, (2022) as reference (lines 251- 265).

We hope that we have addressed Reviewer's perplexities and the manuscript will be now suitable for publication.

Bests

Round 4

Reviewer 2 Report

Comments and Suggestions for Authors

The authors have paid due diligence in revising their manuscript.